# Deep Fisher Networks
# for Large-Scale Image Classification

**Karen Simonyan**          **Andrea Vedaldi**          **Andrew Zisserman**
Visual Geometry Group, University of Oxford
{karen,vedaldi,az}@robots.ox.ac.uk

## Abstract

As massively parallel computations have become broadly available with modern GPUs, deep architectures trained on very large datasets have risen in popularity. Discriminatively trained convolutional neural networks, in particular, were recently shown to yield state-of-the-art performance in challenging image classification benchmarks such as ImageNet. However, elements of these architectures are similar to standard hand-crafted representations used in computer vision. In this paper, we explore the extent of this analogy, proposing a version of the state-of-the-art Fisher vector image encoding that can be stacked in multiple layers. This architecture significantly improves on standard Fisher vectors, and obtains competitive results with deep convolutional networks at a smaller computational learning cost. Our hybrid architecture allows us to assess how the performance of a conventional hand-crafted image classification pipeline changes with increased depth. We also show that convolutional networks and Fisher vector encodings are complementary in the sense that their combination further improves the accuracy.

## 1   Introduction

Discriminatively trained deep convolutional neural networks (CNN) [18] have recently achieved impressive state of the art results over a number of areas, including, in particular, the visual recognition of categories in the ImageNet Large-Scale Visual Recognition Challenge [4]. This success is built on many years of tuning and incorporating ideas into CNNs in order to improve their performance. Many of the key ideas in CNN have now been absorbed into features proposed in the computer vision literature – some have been discovered independently and others have been overtly borrowed. For example: the importance of whitening [11]; max pooling and sparse coding [26, 33]; non-linearity and normalization [20]. Indeed, several standard features and pipelines in computer vision, such as SIFT [19] and a spatial pyramid on Bag of visual Words (BoW) [16] can be seen as corresponding to layers of a standard CNN. However, image classification pipelines used in the computer vision literature are still generally quite shallow: either a global feature vector is computed over an image, and used directly for classification; or, in a few cases, a two layer hierarchy is used, where the outputs of a number of classifiers form the global feature vector for the image (e.g. attributes and classemes [15, 30]).

The question we address in this paper is whether it is possible to improve the performance of off-the-shelf computer vision features by organising them into a deeper architecture. To this end we make the following contributions: (i) we introduce a *Fisher Vector Layer*, which is a generalization of the standard FV to a layer architecture suitable for stacking; (ii) we demonstrate that by discriminatively training several such layers and stacking them into a *Fisher Vector Network*, an accuracy competitive with the deep CNN can be achieved, whilst staying in the realms of conventional SIFT and colour features and FV encodings; and (iii) we show that class posteriors, computed by the deep CNN and FV, are complementary and can be combined to significantly improve the accuracy.

The rest of the paper is organised as follows. After a discussion of the related work, we begin with a brief description of the conventional FV encoding [20] (Sect. 2). We then show how this

representation can be modified to be used as a layer in a deeper architecture (Sect. 3) and how the latter can be discriminatively learnt to yield a deep Fisher network (Sect. 4). After discussing important details of the implementation (Sect. 5), we evaluate our architecture on the ImageNet image classification benchmark (Sect. 6).

**Related work.** There is a vast literature on large-scale image classification, which we briefly review here. One widely used approach is to extract local features such as SIFT [19] densely from each image, aggregate and encode them as high-dimensional vectors, and feed the latter to a classifier, e.g. an SVM. There exists a large variety of different encodings that can be used for this purpose, including the BoW [9, 29] encoding, sparse coding [33], and the FV encoding [20]. Since FV was shown to outperform other encodings [6] and achieve very good performance on various image recognition benchmarks [21, 28], we use it as the basis of our framework. We note that other recently proposed encodings (e.g. [5]) can be readily employed in the place of FV. Most encodings are designed to disregard the spatial location of features in order to be invariant to image transformations; in practice, however, retaining weak spatial information yields an improved classification performance. This can be incorporated by dividing the image into regions, encoding each of them individually, and stacking the result in a composite higher-dimensional code, known as a spatial pyramid [16]. The alternative, which does not increase the encoding dimensionality, is to augment the local features with their spatial coordinates [24].

Another vast family of image classification techniques is based on Deep Neural Networks (DNN), which are inspired by the layered structure of the visual cortex in mammals [22]. DNNs can be trained greedily, in a layer-by-layer manner, as in Restricted Boltzmann Machines [12] and (sparse) auto-encoders [3, 17], or by learning all layers simultaneously, which is relatively efficient if the layers are convolutional [18]. In particular, the advent of massively-parallel GPUs has recently made it possible to train deep convolutional networks on a large scale with excellent performance [7, 14]. It was also shown that techniques such as training and test data augmentation, as well as averaging the outputs of independently trained DNNs, can significantly improve the accuracy.

There have been attempts to bridge these two families, exploring the trade-offs between network depth and width, as well as the complexity of the layers. For instance, dense feature encoding using the bag of visual words was considered as a single layer of a deep network in [1, 8, 32].

## 2   Fisher vector encoding for image classification

The Fisher vector encoding $\phi$ of a set of features $\{x_p\}$ (e.g. densely computed SIFT features) is based on fitting a parametric generative model, e.g. the Gaussian Mixture Model (GMM), to the features, and then encoding the derivatives of the log-likelihood of the model with respect to its parameters [13]. In the particular case of GMMs with diagonal covariances, used here, this leads to the representation which captures the average first and second order differences between the features and each of the GMM centres [20]:

$$\Phi_k^{(1)} = \frac{1}{N\sqrt{\pi_k}} \sum_{p=1}^N \alpha_k(x_p) \left( \frac{x_p - \mu_k}{\sigma_k} \right), \quad \Phi_k^{(2)} = \frac{1}{N\sqrt{2\pi_k}} \sum_{p=1}^N \alpha_k(x_p) \left( \frac{(x_p - \mu_k)^2}{\sigma_k^2} - 1 \right) \quad (1)$$

Here, $\{\pi_k, \mu_k, \sigma_k\}_k$ are the mixture weights, means, and diagonal covariances of the GMM, which is computed on the training set and used for the description of all images; $\alpha_k(x_p)$ is the soft assignment weight of the $p$-th feature $x_p$ to the $k$-th Gaussian. An FV is obtained by stacking the differences: $\phi = \left[ \Phi_1^{(1)}, \Phi_1^{(2)}, \ldots, \Phi_K^{(1)}, \Phi_K^{(2)} \right]$. The encoding describes how the distribution of features of a particular image differs from the distribution fitted to the features of all training images. To make the features amenable to the FV description based on the diagonal-covariance GMM, they are first decorrelated by PCA.

The FV dimensionality is $2Kd$, where $K$ is the codebook size (the number of Gaussians in the GMM), and $d$ is the dimensionality of the encoded feature vector. For instance, FV encoding of a SIFT feature ($d = 128$) using a small GMM codebook ($K = 256$) is $65.5K$-dimensional. This means that high-dimensional feature encodings can be quickly computed using small codebooks. Using the same codebook size, BoW and sparse coding are only $K$-dimensional and less discriminative, as demonstrated in [6]. From another point of view, given the desired encoding dimensionality, these methods would require $2d$-times larger codebooks than needed for FV, which would lead to impractical computation times.

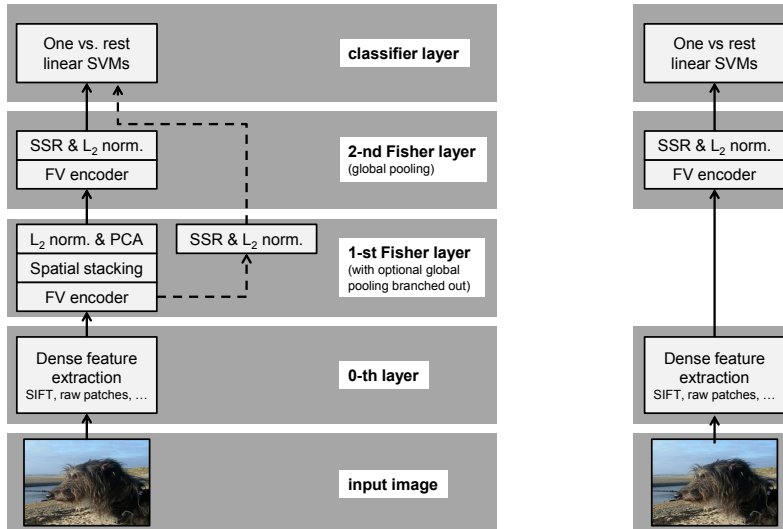

Figure 1: **Left: Fisher network (Sect. 4) with two Fisher layers. Right: conventional pipeline using a shallow Fisher vector encoding.** As shown in Sect. 6, making the conventional pipeline slightly deeper by injecting a single Fisher layer substantially improves the classification accuracy.

As can be seen from (1), the (unnormalised) FV encoding is additive with respect to image features, i.e. the encoding of an image is an average of the individual encodings of its features. Following [20], FV performance is further improved by passing it through Signed Square-Rooting (SSR) and $L_2$ normalisation. Finally, the high-dimensional FV is usually coupled with a one-vs-rest linear SVM classifier, and together they form a conventional image classification pipeline [21] (see Fig. 1), which serves as a baseline for our classification framework.

## 3  Fisher layer

The conventional FV representation of an image (Sect. 2), effectively encodes each local feature (e.g. SIFT) into a high-dimensional representation, and then aggregates these encodings into a single vector by global sum-pooling over the whole image (followed by normalisation). This means that the representation describes the image in terms of the local patch features, and can not capture more complex image structures. Deep neural networks are able to model the feature hierarchies by passing an output of one feature computation layer as the input to the next one. We adopt a similar approach here, and devise a feed-forward feature encoding layer (which we term a *Fisher layer*), which is based on off-the-shelf Fisher vector encoding. The layers can then be stacked into a deep network, which we call a *Fisher network*.

The architecture of the $l$-th Fisher layer is depicted in Fig. 2. On the input, it receives $d_l$-dimensional features ($d_l \sim 10^2$), densely computed over multiple scales on a regular image grid. The features are assumed to be decorrelated using PCA. The layer then performs feed-forward feature transformation in three sub-layers.

The first one computes *semi-local* FV encodings by pooling the input features not from the whole image, but from a dense set of semi-local regions. The resulting FVs form a new set of densely sampled features that are more discriminative than the input ones and less local, as they integrate information from larger image areas. The FV encoder (Sect. 2) uses a layer-specific GMM with $K_l$ components, so the dimensionality of each FV is $2K_l d_l$, which, considering that FVs are computed densely, might be too large for practical applications. Therefore, we decrease FV dimensionality by projection onto $h_l$-dimensional subspace using a *discriminatively trained* linear projection $W_l \in \mathcal{R}^{h_l \times 2K_l d_l}$. In practice, this is carried out using an efficient variant of the FV encoder (Sect. 5). In the second sub-layer, the spatially adjacent features are stacked in a $2 \times 2$ window, which produces $4h_l$-dimensional dense feature representation. Finally, the features are $L_2$-normalised and PCA-projected to $d_{l+1}$-dimensional subspace using the linear projection $U_l \in \mathcal{R}^{d_{l+1} \times 4h_l}$, and passed as the input to the $(l+1)$-th layer. Each sub-layer is explained in more detail below.

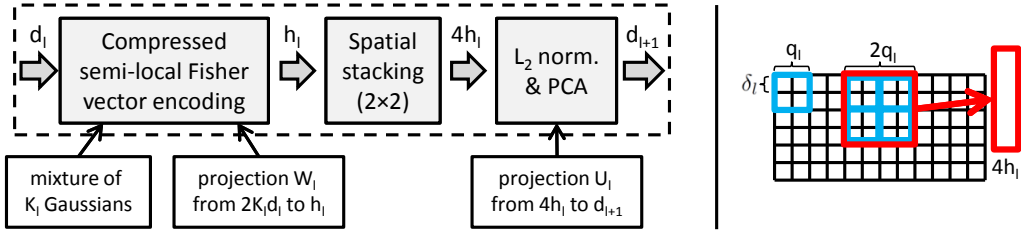

Figure 2: **The architecture of a single Fisher layer.** *Left:* the arrows illustrate the data flow through the layer; the dimensionality of densely computed features is shown next to the arrows. *Right:* spatial pooling (the blue squares) and stacking (the red square) in sub-layers 1 and 2 respectively.

**Fisher vector pooling (sub-layer 1).** The key idea behind the first sub-layer is to aggregate the FVs of individual features over a family of semi-local spatial neighbourhoods. These neighbourhoods are overlapping square regions of size $q_l \times q_l$, sampled every $\delta_l$ pixels (see Fig. 2); compared to the regions used in global or spatial pyramid pooling [20], these are smaller and sampled much more densely. As a result, instead of a single FV, describing the whole image, the image is represented by a large number of densely computed semi-local FVs, each of which describes a spatially adjacent set of local features, computed by the previous layer. Thus, the new feature representation can capture more complex image statistics with larger spatial support. We note that due to additivity, computing the FV of a spatial neighbourhood corresponds to the sum-pooling over the neighbourhood, a stage widely used in DNNs. The high dimensionality of Fisher vectors, however, brings up the computational complexity issue, as storing and processing thousands of dense FVs per image (each of which is $2K_l d_l$-dimensional) is prohibitive at large scale. We tackle this problem by employing discriminative dimensionality reduction for high-dimensional FVs, which makes the layer learning procedure *supervised*. The dimensionality reduction is carried out using a linear projection $W_l$ onto an $h_l$-dimensional subspace. As will be shown in Sect. 5, compressed FVs can be computed very efficiently without the need to compute the full-dimensional FVs first, and then project them down.

A similar approach (passing the output of a feature encoder to another encoder) has been previously employed by [1, 8, 32], but in their case they used bag-of-words or sparse coding representations. As noted in [8], such encodings require large codebooks to produce discriminative feature representations. This, in turn, makes these approaches hardly applicable to the datasets of ImageNet scale [4]. As explained in Sect. 2, FV encoders do not require large codebooks, and by employing supervised dimensionality reduction, we can preserve the discriminative ability of FV even after the projection onto a low-dimensional space, similarly to [10].

**Spatial stacking (sub-layer 2).** After the dimensionality-reduced FV pooling (Sect. 3), an image is represented as a spatially dense set of low-dimensional multi-scale discriminative features. It should be noted that local sum-pooling, while making the representation invariant to small translations, is agnostic to the relative location of aggregated features. To capture the spatial structure within each feature's neighbourhood, we incorporate the stacking sub-layer, which concatenates the spatially adjacent features in a $2 \times 2$ window (Fig. 2). This step is similar to $4 \times 4$ stacking employed in SIFT.

**Normalisation and PCA projection (sub-layer 3).** After stacking, the features are $L_2$-normalised, which improves their invariance properties. This procedure is closely related to Local Contrast Normalisation, widely used in DNNs. Finally, before passing the features to the FV encoder of the next layer, PCA dimensionality reduction is carried out, which serves two purposes: (i) features are decorrelated so that they can be modelled using diagonal-covariance GMMs of the next layer; (ii) dimensionality is reduced from $4h_l$ to $d_{l+1}$ to keep the image representation compact and the computational complexity limited.

**Multi-scale computation.** In practice, the Fisher layer computation is repeated at multiple scales by changing the pooling window size $q_l$ (the PCA projection in sub-layer 3 is the same for all scales). This allows a single layer to capture multi-scale statistics, which is different from typical DNN architectures, which use a single pooling window size per layer. The resulting dense multi-scale features, computed by the layer, form the input of the next layer (similarly to the dense multi-scale SIFT features). In Sect. 6 we show that a multi-scale Fisher layer indeed brings an improvement, compared to a fixed pooling window size.

# 4 Fisher network

Our image classification pipeline, which we coin *Fisher network* (shown in Fig. 1) is constructed by stacking several (at least one) Fisher layers (Sect. 3) on top of dense features, such as SIFT or raw image patches. The penultimate layer, which computes a single-vector image representation, is the special case of the Fisher layer, where sum-pooling is only performed globally over the whole image. We call this layer the *global* Fisher layer, and it effectively computes a full-dimensional normalised Fisher vector encoding (the dimensionality reduction stage is omitted since the computed FV is directly used for classification). The final layer is an off-the-shelf ensemble of one-vs-rest binary linear SVMs. As can be seen, a Fisher network generalises the standard FV pipeline of [20], as the latter corresponds to the network with a single global Fisher layer.

**Multi-layer image descriptor.** Each subsequent Fisher layer is designed to capture more complex, higher-level image statistics, but the competitive performance of shallow FV-based frameworks [21] suggests that low-level SIFT features are already discriminative enough to distinguish between a number of image classes. To fully exploit the hierarchy of Fisher layers, we branch out a globally pooled, normalised FV from each of the Fisher layers, not just the last one. These image representations are then concatenated to produce a rich, multi-layer image descriptor. A similar approach has previously been applied to convolutional networks in [25].

## 4.1 Learning

The Fisher network is trained in a supervised manner, since each Fisher layer (apart from the global layer) depends on discriminative dimensionality reduction. The network is trained greedily, layer by layer. Here we discuss how the (non-global) Fisher layer can be efficiently trained in the large-scale scenario, and introduce two options for the projection learning objective.

**Projection learning proxy.** As explained in Sect. 3, we need to learn a discriminative projection $W$ to significantly reduce the dimensionality of the densely-computed semi-local FVs. At the same time, the only annotation available for discriminative learning in our case is the class label of the whole image. We exploit this information by requiring that projected semi-local FVs are good predictors of the image class. Taking into account that (i) it may be unreasonable to require *all* local feature occurrences to predict the object class (the support of some features may not even cover the object), and (ii) there are too many features to use all of them in learning ($\sim 10^4$ semi-local FVs for each of the $\sim 10^6$ training images), we optimize the *average* class prediction of all the features in a layer, rather than the prediction of individual feature occurrences.

In particular, we construct a learning proxy by computing the average $\psi$ of all unnormalised, unprojected semi-local FVs $\phi_s$ of an image, $\psi = \frac{1}{S}\sum_{s=1}^{S}\phi_s$, and defining the learning constraints on $\psi$ using the image label. Considering that $W\psi = \frac{1}{S}\sum_{s=1}^{S}W\phi_s$, the projection $W$, learnt for $\psi$, is also applicable to individual semi-local FVs $\phi_s$. The advantages of the proxy are that the image-level class annotation can now be utilised, and during projection learning we only need to store a single vector $\psi$ per image. In the sequel, we define two options for the projection learning objective, which are then compared in Sect. 6.

**Bi-convex max-margin projection learning.** One approach to discriminative dimensionality reduction learning consists in finding the projection onto a subspace where the image classes are as linearly separable as possible [10, 31]. This corresponds to the bilinear class scoring function: $v_c^T W\psi$, where $W$ is the linear projection which we seek to optimise and $v_c$ is the linear model (e.g. an SVM) of the class $c$ in the projected space. The max-margin optimisation problem for $W$ and the ensemble $\{v_c\}$ takes the following form:

$$\sum_i \sum_{c' \neq c(i)} \max\left[\left(v_{c'} - v_{c(i)}\right)^T W\psi_i + 1, 0\right] + \frac{\lambda}{2}\sum_c \|v_c\|_2^2 + \frac{\mu}{2}\|W\|_F^2, \qquad (2)$$

where $c_i$ is the ground-truth class of an image $i$, $\lambda$ and $\mu$ are the regularisation constants. The learning objective is bi-convex in $W$ and $v_c$, and a local optimum can be found by alternation between the convex problems for $W$ and $\{v_c\}$, both of which can be solved in primal using a stochastic sub-gradient method [27]. We initialise the alternation by setting $W$ to the PCA-whitening matrix $W_0$. Once the optimisation has converged, the classifiers $v_c$ are discarded, and we keep the projection $W$.

**Projection onto the space of classifier scores.** Another dimensionality reduction technique, which we consider in this work, is to train one-vs-rest SVM classifiers $\{u_c\}_{c=1}^{C}$ on the full-dimensional FVs $\psi$, and then use the $C$-dimensional vector of SVM outputs as the compressed representation of $\psi$. This corresponds to setting the $c$-th row of the projection matrix $W$ to the SVM model $u_c$. This approach is closely related to attribute-based representations and classemes [15, 30], but in our case we do not use any additional data annotated with a different set of (attribute) classes to train the models; instead, the $C = 1000$ classifiers trained directly on the ILSVRC dataset are used. If a specific target dimensionality is required, PCA dimensionality reduction can be further applied to the classifier scores [10], but in our case we applied PCA after spatial stacking (Sect. 3).

The advantage of using SVM models for dimensionality reduction is, mostly, computational. As we will show in Sect. 6, both formulations exhibit a similar level of performance, but training $C$ one-vs-rest classifiers is much faster than performing alternation between SVM learning and projection learning in (2). The reason is that one-vs-rest SVM training can be easily parallelised, while projection learning is significantly slower even when using a parallel gradient descent implementation.

## 5   Implementation details

**Efficient computation of hard-assignment Fisher vectors.** In the original FV encoding formulation (1), each feature is soft-assigned to all $K$ Gaussians of the GMM by computing the assignment weights $\alpha_k(x_p)$ as the responsibilities of GMM component $k$ for feature $p$: $\alpha_k(x_p) = \frac{\pi_k \mathcal{N}_k(x_p)}{\sum_j \pi_j \mathcal{N}_j(x_p)}$, where $\mathcal{N}_k(x_p)$ is the likelihood of $k$-th Gaussian. To facilitate an efficient computation of a large number of dense FVs per image, we introduce and utilise a fast variant of FV (which we term hard-FV), which uses hard assignments of features to Gaussians, computed as

$$\alpha_k(x_p) = \begin{cases} 1 & \text{if } k = \arg\max_j \pi_j \, \mathcal{N}_j(x_p) \\ 0 & \text{otherwise} \end{cases} \tag{3}$$

Hard-FVs are inherently sparse; this allows for the fast computation of projected FVs $W_l \phi$. Indeed, it is easy to show that $W_l \phi = \sum_{k=1}^{K} \sum_{p \in \Omega_k} \left( W_l^{(k,1)} \Phi_k^{(1)}(p) + W_l^{(k,2)} \Phi_k^{(2)}(p) \right)$, where $\Omega_k$ is the set of input vectors hard-assigned to the GMM component $k$, and $W_l^{(k,1)}, W_l^{(k,2)}$ are the sub-matrices of $W_l$ which correspond to the 1st and 2nd order differences $\Phi_k^{(1),(2)}(p)$ between the feature $x_p$ and the $k$-th GMM mean (1). This suggests the fast computation procedure: each $d_l$-dimensional input feature $x_p$ is first hard-assigned to a Gaussian $k$ based on (3). Then, the corresponding $d_l$-D differences $\Phi_k^{(1),(2)}(p)$ are computed and projected using small $h_l \times d_l$ sub-matrices $W_l^{(k,1)}, W_l^{(k,2)}$, which is fast. The algorithm avoids computing high-dimensional FVs, followed by the projection using a large matrix $W_l \in \mathcal{R}^{h_l \times 2K_l d_l}$, which is prohibitive since the number of dense FVs is high.

**Implementation.** Our SIFT feature extraction follows that of [21]. Images are rescaled so that the number of pixels is 100K. Dense RootSIFT [2] is computed on $24 \times 24$ patches over 5 scales (scale factor $\sqrt[3]{2}$) with a 3 pixel step. We also employ SIFT augmentation with the patch spatial coordinates [24]. During training, high-dimensional FVs, computed by the 2nd Fisher layer, are compressed using product quantisation [23]. The learning framework is implemented in Matlab, speeded up with C++ MEX. The computation is carried out on CPU without the use of GPU. Training the Fisher network on top of SIFT descriptors on 1.2M images of ILSVRC-2010 [4] dataset takes about one day on a 200-core cluster. Image classification time is $\sim 2$s on a single core.

## 6   Evaluation

In this section, we evaluate the proposed Fisher network on the dataset, introduced for the ImageNet Large Scale Visual Recognition Challenge (ILSVRC) 2010 [4]. It contains images of 1000 categories, with 1.2M images available for training, 50K for validation, and 150K for testing. Following the standard evaluation protocol for the dataset, we report both top-1 and top-5 accuracy (%) computed on the test set. Sect. 6.1 evaluates variants of the Fisher network on a subset of ILSVRC to identify the best one. Then, Sect. 6.2 evaluates the complete framework.

### 6.1   Fisher network variants

We begin with comparing the performance of the Fisher network under different settings. The comparison is carried out on a subset of ILSVRC, which was obtained by random sampling of 200

Table 1: **Evaluation of dimensionality reduction, stacking, and normalisation sub-layers on a 200 class subset of ILSVRC-2010.** The following configuration of Fisher layers was used: $d_1 = 128$, $K_1 = 256$, $q_1 = 5$, $\delta_1 = 1$, $h_1 = 200$ (number of classes), $d_2 = 200$, $K_2 = 256$. The baseline performance of a shallow FV encoding is 57.03% and 78.9% (top-1 and top-5 accuracy).

| dim-ty reduction | stacking | L2 norm-n | top-1 | top-5 |
|---|---|---|---|---|
| classifier scores | | ✓ | 59.69 | 80.29 |
| classifier scores | ✓ | | 59.42 | 80.44 |
| classifier scores | ✓ | ✓ | **60.22** | 80.93 |
| bi-convex | ✓ | ✓ | 59.49 | **81.11** |

Table 2: **Evaluation of multi-scale pooling and multi-layer image description on the subset of ILSVRC-2010.** The following configuration of Fisher layers was used: $d_1 = 128$, $K_1 = 256$, $h_1 = 200$, $d_2 = 200$, $K_2 = 256$. Both Fisher layers used spatial coordinate augmentation. The baseline performance of a shallow FV encoding is 59.51% and 80.50% (top-1 and top-5 accuracy).

| pooling window size $q_1$ | pooling stride $\delta_1$ | multi-layer | top-1 | top-5 |
|---|---|---|---|---|
| 5 | 1 | | 61.56 | 82.21 |
| $\{5, 7, 9, 11\}$ | 2 | | 62.16 | 82.43 |
| $\{5, 7, 9, 11\}$ | 2 | ✓ | **63.79** | **83.73** |

classes out of 1000. To avoid over-fitting indirectly on the test set, comparisons in this section are carried on the validation set. In our experiments, we used SIFT as the first layer of the network, followed by two Fisher layers (the second one is global, as explained in Sect. 4).

**Dimensionality reduction, stacking, and normalisation.** Here we quantitatively assess the three sub-layers of a Fisher layer (Sect. 3). We compare the two proposed dimensionality reduction learning schemes (bi-convex learning and classifier scores), and also demonstrate the importance of spatial stacking and $L_2$ normalisation. The results are shown in Table 1. As can be seen, both spatial stacking and $L_2$ normalisation improve the performance, and dimensionality reduction via projection onto the space of SVM classifier scores performs on par with the projection learnt using the bi-convex formulation (2). In the following experiments we used the classifier scores for dimensionality reduction, since their training can be parallelised and is significantly faster.

**Multi-scale pooling and multi-layer image representation.** In this experiment, we compare the performance of semi-local FV pooling using single and multiple window sizes (Sect. 3), as well as single- and multi-layer image representations (Sect. 4). From Table 2 it is clear that using multiple pooling window sizes is beneficial compared to a single window size. When using multi-scale pooling, the pooling stride was increased to keep the number of pooled semi-local FVs roughly the same. Also, the multi-layer image descriptor obtained by stacking globally pooled and normalised FVs, computed by the two Fisher layers, outperforms each of these FVs taken separately. We also note that in this experiment, unlike the previous one, both Fisher layers utilized spatial coordinate augmentation of the input features, which leads to a noticeable boost in the shallow baseline performance (from 78.9% to 80.50% top-5 accuracy). Apart from our Fisher network, multi-scale pooling can be readily employed in convolutional networks.

### 6.2 Evaluation on ILSVRC-2010

Now that we have evaluated various Fisher layer configurations on a subset of ILSVRC, we assess the performance of our framework on the full ILSVRC-2010 dataset. We use off-the-shelf SIFT and colour features [20] in the feature extraction layer, and demonstrate that significant improvements can be achieved by injecting a single Fisher layer into the conventional FV-based pipeline [23].

The following configuration of Fisher layers was used: $d_1 = 80$, $K_1 = 512$, $q_1 = \{5, 7, 9, 11\}$, $\delta_1 = 2$, $h_1 = 1000$, $d_2 = 256$, $K_2 = 256$. On both Fisher layers, we used spatial coordinate augmentation of the input features. The first Fisher layer uses a large number of GMM components $K_l$, since it was found to be beneficial for shallow FV encodings [23], used here as a baseline. The one-vs-rest SVM scores were Platt-calibrated on the validation set (we did not use calibration for semi-local FV dimensionality reduction).

The results are shown in Table 3. First, we note that the globally pooled Fisher vector, branched out of the first Fisher layer (which effectively corresponds to the conventional FV encoding [23]), results in better accuracy than reported in [23], which validates our implementation. Using the 2nd Fisher layer on top of the 1st one leads to a significant performance improvement. Finally, stacking the FVs, produced by the 1st and 2nd Fisher layers, pushes the accuracy even further.

Table 3: **Performance on ILSVRC-2010 using dense SIFT and colour features.** We also specify the dimensionality of SIFT-based image representations.

| pipeline setting | SIFT only | | | SIFT & colour | |
|---|---|---|---|---|---|
| | dimension | top-1 | top-5 | top-1 | top-5 |
| 1st Fisher layer | 82K | 46.52 | 68.45 | 55.35 | 76.35 |
| 2nd Fisher layer | 131K | 48.54 | 71.35 | 56.20 | 77.68 |
| multi-layer (1st and 2nd Fisher layers) | 213K | **52.57** | **73.68** | **59.47** | **79.20** |
| Sánchez et al. [23] | 524K | N/A | 67.9 | 54.3 | 74.3 |

The state of the art on the ILSVRC-2010 dataset was obtained using an 8-layer convolutional network [14], i.e. twice as deep as the Fisher network considered here. Using training and test set augmentation based on jittering (not employed here), they achieved the top-1 / top-5 accuracy of 62.5% / 83.0%. Without test set augmentation (i.e. using only the original images for class scoring), their result is 61% / 81.7%. In our case, we did not augment neither the training, nor the test set, and achieved 59.5% / 79.2%. For reference, our baseline shallow FV accuracy is 55.4% / 76.4%. We conclude that injecting a single intermediate layer leads to a significant performance boost (+4.1% top-1 accuracy), but deep CNNs are still somewhat better (+1.5% top-1 accuracy). These results are however quite encouraging since they were obtained by using off-the-shelf features and encodings, reconfigured to add a single intermediate layer. Notably, our model did not require an optimised GPU implementation, nor it was necessary to control over-fitting by techniques such as dropout [14] and training set augmentation.

Finally, we demonstrate that the Fisher network and deep CNN representations are complementary by combining the class posteriors obtained from CNN with those of a Fisher network. To this end, we re-implemented the deep CNN of [14] using their publicly available cuda-convnet toolbox. Our implementation performs slightly better, giving 62.91% / 83.19% (with test set augmentation). The multiplication of CNN and Fisher network posteriors leads to a significantly improved accuracy: **66.75% / 85.64%**. It should be noted that another way of improving the CNN accuracy, used in [14] on ImageNet-2012 dataset, consists in training several CNNs and averaging their posteriors. Further study of the complementarity of various deep and shallow representations is beyond the scope of this paper, and will be addressed in future research.

## 7 Conclusion

We have shown that Fisher vectors, a standard image encoding method, are amenable to be stacked in multiple layers, in analogy to the state-of-the-art deep neural network architectures. Adding a single layer is in fact sufficient to significantly boost the performance of these shallow image encodings, bringing their performance closer to the state of the art in the large-scale classification scenario [14]. The fact that off-the-shelf image representations can be simply and successfully stacked indicates that deep schemes may extend well beyond neural networks.

## Acknowledgements

This work was supported by ERC grant VisRec no. 228180.

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
