[Reviews · NeurIPS 2013]

Submitted by Assigned_Reviewer_7

The authors propose a `deep' extension for the Fischer vector encoding by stacking multiple layers, based on various training, pooling and normalization operations. Several innovations are introduced including a Fischer vector layer as well as various unsupervised and supervised training regimes. Experiments show that a non-shallow architecture based on Fischer encoding indeed improves over the flat counterpart. Improvements are significant. The paper is clearly presented and discusses the design choices made as well as some of the limitations in a balanced manner. The paper builds upon some of the deep architecture approaches, which in contrast to CNN, are not fully trained, but rely on some of the computer vision success stories in designing local feature descriptors (SIFT, Fischer, etc.). Developing some of the computer vision descriptors for usage within a deep architecture looks significant enough to me.


Semantic segmentation with second-order pooling, ECCV 2012 uses second-order statistics with good results for semantic segmentation and image classification. Could such descriptors be used within your multi-layer framework? It may be advantageous as these are smaller dimensionality than Fischer. The encoding is explicit, can be used with linear predictors, but corresponds to a non-linear kernel.
Summary: A plausible multilayer extension to Fischer encoding, with state of the art results in Imagenet. Clear presentation, sensible techniques, relevant experiments.

Submitted by Assigned_Reviewer_9

The paper proposes a new image representation for recognition based on a stacking of two layers of Fisher vector encoders, with the first layer capturing semi-local information and the second performing sum-pooling aggregation over the entire picture. The approach is inspired by the recent success of deep convolutional networks (CNN). The key-difference is that the architecture proposed in this paper is predominantly hand-designed with relatively few parameters learned compared to CNNs. This is both the strength and the weakness of the approach as it leads to much faster training but also slighter lower accuracy compared to fully learned deep networks.

The high-level idea of the approach is well presented and is easy to grasp. However, on the other hand the specific model details are difficult to comprehend: I had to go over the key sections multiple times and certain aspects of the implementation are still a bit unclear in my mind. For example, are the Fisher encoding parameters (mean and covariance) of the different semi-local areas distinct or are they shared? Also, I think it would be beneficial to spell out explicitly the input/output dimensionality of each layer in terms of the essential parameters (# pixels, # multi-layers, pooling window size and stride, etc.). Hellinger map is only mentioned in the figure but not described in the text.

In my view this is one of the first attempts at bridging the trend of hand-engineered image representations traditionally used in computer vision and the fully-learned approach of deep networks. Such a study was long due in my opinion. However, at the same time I am a bit disappointed that the study is focused only on one specific type of hand-engineered feature (Fisher vector) and it is quite superficial in the empirical evaluation, e.g., why not studying deeper (> 2 layers) stacks of Fisher layers progressively pooling information over larger and larger regions? The use of Fisher vectors as feature of choice is justified in the paper by its good performance and by the need of high-dimensional vectors for accurate recognition. Yet, dimensionality reduction is then employed to avoid having an explosion in the number of parameters and to maintain the method practically feasible. Given this compromise, perhaps alternative lower-dimensional features could have been considered.

In principle, an important advantage over vanilla Fisher encoding is that the proposed approach considers and analyzes local neighborhood information before the final global aggregation over the entire image. However, in order to gauge the effective improvement, it would have been interesting to compare the method to the version of Fisher encoding using the spatial pyramid (i.e., [10]).
Summary: The paper proposes an interesting hybrid between mainstream hand-engineered features and fully-learned hierarchical models. Experimentally, the proposed model is shown to perform slightly better than the well-established Fisher vector but not as well as deep networks.

Submitted by Assigned_Reviewer_11

This paper uses Fisher Vectors as inner building blocks in a recognition architecture. The basic Fisher vector module had previously demonstrated superior performance in recognition application. Here, it is augmented with discriminative linear projection for dimensionality reduction, and multiscale local pooling, to make it suitable for stacking. Inputs of all layers are jointly used for classification. Experiments show convincing performance on ImageNet.
This paper is well organized, well written, and shows good results, establishing that (1) adding another layer of Fisher vector extraction is useful, (2) using both layers jointly is better than using any one of them in isolation (this confirms earlier work, as stated in the paper), (3) using multiscale pooling is better than using a single pooling neighborhood size. Concerning this last point: the idea of multiple pooling scales vs. a single one is analogous to the difference between spatial pyramid and simple bag of words, so this is not really new, and should be mentioned.
A small gripe with the writing is with two ways that the paper is sold: (1) the conclusion reads, "the architecture is largely hand-crafted, and requires lighter training." Why is using hand-crafted features seen as a good thing? The fact that other deep networks incorporate training allows adaptation to other domains. For a vision application, training could be avoided by plugging in previously trained weights, using training samples as weights, using only unsupervised pre-training, etc. -- the option to fully train a network simply results in the best performance, but the amount of needed training is flexible if one is willing to accept traning-performance trade-offs.
(2) the introduction and conclusion take aim at a strawman: e.g. in the introduction, l.48: "are the deep architectures used in CNN really necessary, or can comparable performance be obtained simply by stacking the well crafted features developed in the computer vision literature?". This might unnecessarily alienate the deep learning crowd, given that it has so often been emphasized that all types of modules are suitable for embedding in deep learning architectures, and bits and pieces of computer vision architectures have in fact been used (e.g. see work by Weston and Collobert; the angle taken by Coates et al. in ref. 22; the use of SVM and spatial pyramids mixed with convolutional network layers in Kavukcuoglu et al. NIPS 2010.) Instead, a better way to describe the contribution of the paper could be to reverse the framing and ask whether conventional computer vision features could benefit from the stacking principles used in deep architectures.
Minor comments: Ref. 7 used for instances of sparse coding and max pooling should be replaced by earlier work, e.g.:
Yang et al. CVPR 2009,
Linear spatial pyramid matching using sparse coding for image classification
Summary: This is a strong, technically sound paper with an intuitively appealing approach to transforming a simple module for stacking, and good experimental results. Writing and presentation are clear, and offer a good demonstration that conventional feature extraction modules in vision can benefit from stacking.
Author Feedback

Author rebuttal: 